# Learning Semantically Meaningful Representations Through Embodiment

## Abstract

How do humans acquire a meaningful understanding of the world with little to no supervision or semantic labels provided by the environment? Here we investigate embodiment and a closed loop between action and perception as one key component in this process. We take a close look at the representations learned by a deep reinforcement learning agent that is trained with visual and vector observations collected in a 3D environment with sparse rewards. We show that this agent learns semantically meaningful and stable representations of its environment without receiving any semantic labels. Our results show that the agent learns to represent the action relevant information extracted from pixel input in a wide variety of sparse activation patterns. The quality of the representations learned shows the strength of embodied learning and its advantages over fully supervised approaches with regards to robustness and generalizability.

## 1 Introduction

When the way supervised neural networks learn is compared to the way humans learn one can easily make out some major differences. Two of those differences are *supervision* and *embodiment*. Taking the example of object recognition from visual observations, a neural network will be presented with thousands of images of the object in question, each of them accompanied by a class label. A toddler in comparison will also collect many observations of the object of interest, however, will do so by interacting with the object, looking at it from different perspectives by moving the head or even moving the object (Bambach et al., 2018). This law-governed change in observations conditioned on the movements of the toddler emphasises the importance of embodied cognition (Engel et al., 2013). It will make it possible to recognize the object as a distinct entity, separate from its surroundings and learn a general concept of it. This allows it to robustly recognize the object again even when seen from new perspectives or under different lighting conditions (Smith & Slone, 2017). When the toddler is now told the name of the object, an almost instantaneous association between label and object can be made without the need of thousands of labeled examples (Samuelson & Smith, 2005). This therefore makes a very efficient strategy for learning stable representations of objects.

Fully supervised neural networks have been shown to suffer from shortcomings that humans usually do not exhibit. Szegedy et al. (2013) showed how very small perturbations to an image, undetectable to the human eye, can drastically change the classification accuracy of a neural network. Even simply holding such adversarial examples in front of a camera (Kurakin et al., 2016) or specific natural images (Hendrycks et al., 2019) can have this effect. The networks seem to possess an over-reliance on local image features such as texture and do not consider global features such as the overall shape and outline of an object (Baker et al., 2018). Considering the training circumstances, this effect is unsurprising. The networks are expected to learn the concept of objects solely from pixel values. Without being able to interact with objects or even just looking at them from slightly different perspectives, it is very difficult to figure out basic knowledge such as object and background relationships. We expect that an active exploration of the world would make it possible to learn a more general and robust concept of objects.

Already in 2001 O'Regan & No (2001) argued that even though it is clear that action requires perception, this relation also reverses. Perception and the understanding of what is perceived requires action (Noë, 2005). According to O'Regan & Noë (2001), *"experience is not something that happens in us but is something we do"* (p.99). They argue that an important part of perception is to learn

how actions affect sensations. These sensory motor contingencies help us make sense of our perceptions, predict them and efficiently sample the environment for information (Engel et al., 2013).

In humans, perception is hugely influenced by how we interact with the world (Witt, 2011). Goals and the expected cost to perform actions to achieve a goal influence our perception of physical entities (Proffitt, 2006). Also more abstract processes such as language comprehension are linked to action systems in the brain (Pulvermüller & Fadiga, 2010). We therefore postulate that in order to teach an artificial agent a true understanding of its (simulated) world it needs to be able to interact with the world. This paper will present results from an embodied agent acting in a virtual 3D world and learning an internal representation of its sensory input. The framework of learning by interacting with the world produced a meaningful and action-oriented internal representation of the agents observations, even though no semantic labels were provided.

## 2    RELATED WORK

There is a strong research interest in learning visual structure in an unsupervised way which can for example be approached by using auto-encoders (Tschannen et al., 2018). In its simplest form, relevant structure is supposed to change slowly(Körding & König, 2001; Wiskott & Sejnowski, 2002) facilitating learning of invariant representations. To further incorporate a time component and learn visual structure and changes over time, future frame prediction is a commonly used task (Villegas et al., 2017; Srivastava et al., 2015; Patraucean et al., 2015; Oliu et al., 2017; Mahjourian et al., 2016; Finn et al., 2016; Denton & Birodkar, 2017). However, only a few of the papers dealing with time series prediction actually investigate the learned representations in the network (Lotter et al., 2015; Qiao et al., 2018).

An alternative concept for unsupervised representation learning is the use of predictive coding (Rao & Ballard, 1999) which can be applied to train ANNs (van den Oord et al., 2018; Wen et al., 2018; Han et al., 2018; Lotter et al., 2016). The idea of predicting future observations based on current actions can also be used in a reinforcement learning setting to inject agents with some sense of curiosity (Pathak et al., 2017). Ha & Schmidhuber (2018) has shown that training a recurrent world model using a variational auto-encoder can increase the performance of agents in several games. Chaplot et al. (2019) show how jointly training semantic goal navigation and embodied question answering can improve performance on both of these tasks. Also, simply seeing a visual scene from different angles can get a network to learn disentangled representations of individual objects (Eslami et al., 2018) and get it to imagine the scene from a previously unseen viewpoint (Eslami et al., 2018; Rosenbaum et al., 2018).

Researchers who investigate representation learning in reinforcement learning agents often use additional regularization or losses to enforce a certain representation in the latent space (Nachum et al., 2018; Lesort et al., 2018; de Bruin et al., 2018). Shang et al. (2019) even gets agents to explicitly learn world graphs. As such explicit constraints are biologically implausible, we will investigate what kind of representations arise naturally within an embodied training setup. Lillicrap et al. (2015) have already shown results from a simple deep reinforcement learning agent which indicate that perceptually similar observations are mapped close to each other in the latent space. We will further investigate this and look explicitly at the type of representation encoding that is learned as well as the meaningfulness of the representation and the type of information that is encoded.

## 3    EXPERIMENTS

### 3.1    TRAINING A DEEP REINFORCEMENT LEARNING AGENT

The representation under investigation in this paper is the activation in the hidden layer of a deep neural network trained in a reinforcement learning environment. Figure 1 shows the network structure of the agent. As input, it receives visual and vector observations from the simulated environment with size 168x168x3 and size 8 respectively. The visual and vector observations are first processed separately by two convolutional layers and two dense layers for the visual input and two dense layers for the vector input, until they are concatenated into one encoded state. This encoded state has dimensionality 512 where 256 of its activations come from the visual encoding pathway and 256

from the vector encoding pathway. The encoded state of the high dimensional visual input and its properties will be the main focus of this paper.

Based on the encoded state, two dense layers output action probabilities and a value estimate. The action probabilities are translated into actions and sent to the environment to obtain the next set of observations. They are also used, together with the value estimate and the reward received from the environment to optimize the neural network and thereby, train the agent to perform better actions. For training of the network proximal policy optimization (PPO) is used which is an efficient policy gradient method for deep reinforcement learning (Schulman et al., 2017). More specifically, the PPO implementation and reinforcement learning framework of the Unity ML-Agents toolkit is used (Juliani et al., 2018). The overall setup and training procedure are chosen such that a neural network is trained in an embodied way with a closed loop between perception and action.

As setting for training, the environment proposed in the Unity obstacle tower challenge (Juliani et al., 2019) is used. This environment was introduced as a new benchmark in reinforcement learning for pixel-based learning in a procedurally-generated 3D environment using a sparse reward signal. The agent needs to learn to navigate through a 3-dimensional maze environment, solving successively harder tasks. Every level of the tower consists of several rooms, connected by doors. When reaching the final door on a floor the agent receives a reward of one and is placed on the next, randomly generated floor of the tower. Starting from level five the agent needs to learn picking up a key which unlocks a key door and gives the agent access to further rooms that lead to the next level door. The key can be placed in one of the rooms on the ground or on a static or moving platform. Starting from level ten a new type of door is introduced which opens only after a puzzle is solved. The puzzle requires the agent to push a block onto a colored spot on the ground. The randomly generated floors can be illuminated in different color variations and the visual theme of the environment can vary. As overfitting to specific color values in the input or floor layouts is not useful, this makes it important for the agent to learn general and stable representations.

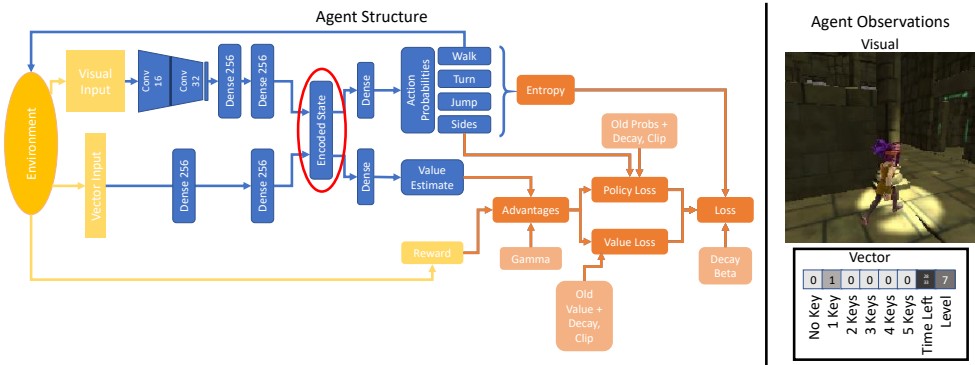

Figure 1: (Left) Network structure of the deep reinforcement learning agent. The environment and everything coming from it *(Yellow)*. The deep neural network, optimized with gradient descent, facilitated by PPO *(Blue)*. Parts of the PPO algorithm used to optimize the neural network *(Orange)*. (Right) Agent observations of one frame, one can see a normal door to the right of the agent. The agent receives visual and vector observations at each frame. Visual observations are of size 168x168x3. Vector observations are of size 8.

The agent observations are collected from a third person view RGB camera. Additionally, a small vector observation is provided indicating the number of keys the agent is holding as well as the time remaining and the current level. Rewards are very sparse as a reward (of value 1) is only received when walking through a final level door or when picking up a key. A small reward of value 0.1 is given when walking through normal doors and when picking up small blue orbs that provide additional time. One episode ends when the agent runs out of time, which means that the better the agent gets the longer exploring the tower is possible as more time-orbs can be collected and extra time received by going through level doors. The actions of the agent are discrete and divided into four action branches. One for moving forward or backwards, one to control the camera rotation, one for jumping and one for moving left or right. The distribution of these actions in a trained agent is show in the appendix (figure 10).

## 3.2 AGENT PERFORMANCE

After training the agent for 30 million steps using the parameters specified in the appendix A.1 level 8 is reached on average. As can be expected from the network structure, the agent never exceeds level ten. The network structure used here incorporates no concept of time such that the agent is unable to solve the puzzles introduced at level 10 as they require some more elaborate planning of a long action sequence.

Figure 2 shows the agent performance during one inference run. This particular run lasted 4000 frames which means the agent saw 4000 observations and performed 4000 actions. As the rewards are very sparse, the agent only received a reward (1 or 0.1) in 76 of these frames which is about 1.9% of all frames (only 15 of

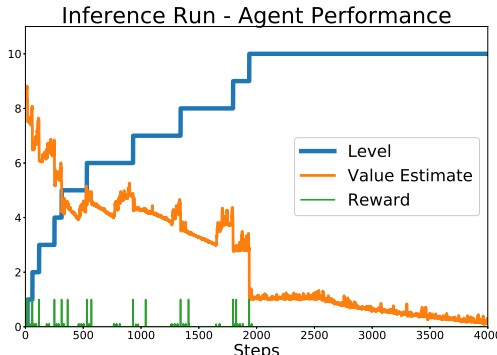

Figure 2: Statistics of one inference run through the tower. The agent reaches level 10 *(blue curve)* and the run ends when the time is up, here after 4000 frames.

those are the full reward of one, the other 61 frames contain a 0.1 reward). One can see how the value estimate, which expresses the reward the agent expects in the future, drops off significantly after the agent reaches level ten as the agent does not expect to solve the puzzle and receive any more rewards. One can also observe how the value estimate rises in the frames leading up to the agent entering a new level. This indicates that the agent recognizes the door to the next level and already anticipates the upcoming reward.

The results of the network trained in the embodied setting will be compared to an untrained network (called random network) as well as the results from an autoencoder. The autoencoder has the exact same structure up to the encoding layer as the trained and random network (see Appendix, figure 8). The autoencoder is trained on a classic frame reconstruction task where its loss is the difference between the network input and output. This training setup separates the factor of embodiment in the training without introducing semantic labels. In figure 9 some example input, outputs and activations in the encoding layer are shown. When comparing the hidden layer activations of the three networks the same 4000 observations of one agent run are used.

## 3.3 SPARSE ACTIVATION PATTERNS

We will take a look at how the activations in the hidden layer of the agent network look like[1]. Figure 3 shows in how many of the 4000 frames of one run each of the 256 neurons in the visual encoding were active. For visualization purposes, the 256x1 vector was reshaped into a grid pattern. The location of a neuron in this grid has no further meaning.

In the visual encoding only 4.64% (mean= 11.88, min=3 max=35 var=17.891) of the neurons are activated in each frame, making it a very sparse activation pattern. Over the investigated total time, 173 of the 256 neurons (=67.58%) are active in at least 1 frame, but only 7 of them (=2.73%) are active in more than 40% of the frames. The most active neuron of the visual embedding is active in 74.15% of all frames. This shows that there is a wide variety in the activation patterns. The agent utilizes most but not all of the available neurons in the

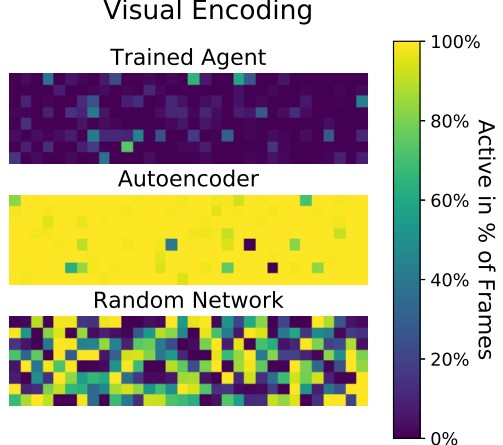

Figure 3: Activation pattern comparison between a trained agent (top), a trained autoencoder (middle) and a random network (bottom). Colors indicate the percentage of frames in which the neuron in the encoded state is active (activation>0).

---

[1]You can find a frame by frame visualization of the activations as well as other interactive displays of our results and our code here: https://embodiedlearning.github.io/ICLR-Submission-2020/

visual embedding at some point but only activates a small part in each frame. In the random network, there are on average 130 active neurons per frame in the visual embedding (min=112 max=152 var=27.352), which is a much denser activity (50.62%). The trained autoencoder has an average of 251 active neurons in each frame (min=227 max=254 var=15.994) and only two of the neurons are never active. Thus, in comparison the trained agent has learned a very sparse representation of the input using selective spikes in activation compared to the continuous activations in the autoencoder encoding information with varying activation strengths.

The sparse representations in the trained agent match observations of sparse encodings for sensory input in insects (Perez-Orive et al., 2002; Laurent, 2002) as well as in the mammalian brain (Young & Yamane, 1992; Brecht & Sakmann, 2002). These form efficient and stable representations of high dimensional sensory input (Olshausen & Field, 2004). The agent picks up on this strategy to efficiently encode input without any explicit regularization being applied. Even though there is no cost associated to using more neurons than needed to encode information, the agent learns to use sparse activation patterns and even leaves some of the available neurons completely unused. The trained agent seems to discover on its own that a sparse representation of the high dimensional image input is more robust and stable to noise (Ahmad & Scheinkman, 2019) which helps to make better action decisions based on this sparse and selective representation. The training setup of the autoencoder enforces a different, more dense way of representing information which is of advantage to optimize its objective but has less resemblance with what we observe in nature.

## 3.4 DISTINCT ACTIVATION PATTERNS

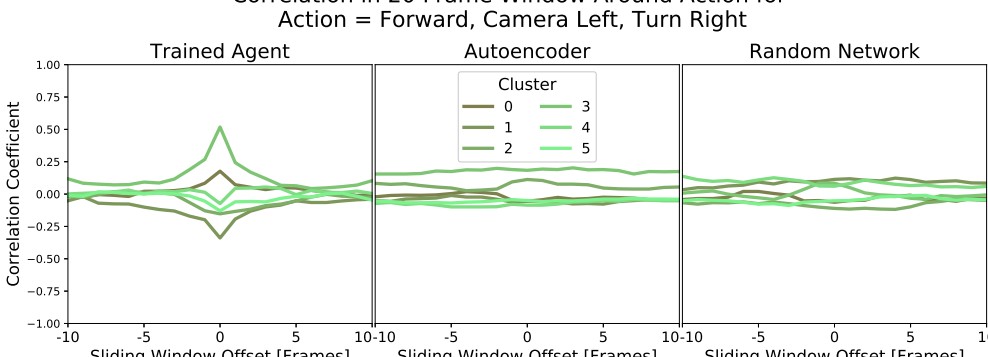

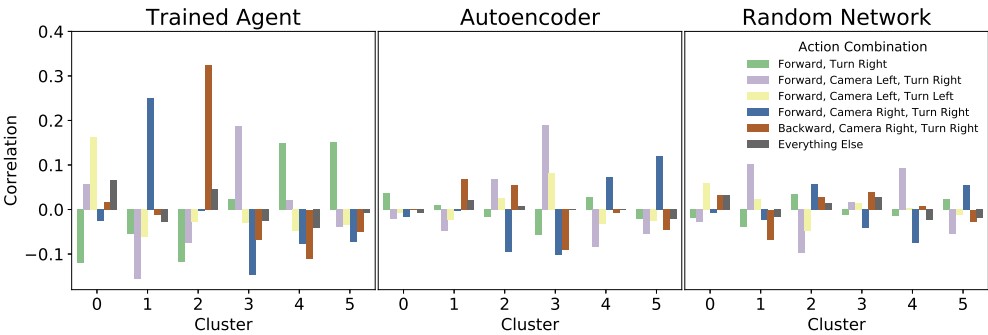

Figure 4: Correlation between the cluster assignment of frames and corresponding action combinations. Clustering performed on the visual embedding of a trained agent (left), an autoencoder (middle) and a random network (right). (Top) Example of correlations for action combination 'Forward + Turn Right' with clusters within a 20-frame window. (Bottom) Bar height represents the average correlation in a 10-frame window around 0 in the top plots for all actions and cluster. It shows distinct correlations between clusters and actions in the trained agent.

To find out if we can discover a general meaningful structure in the activations of the hidden layer, we first perform k-means clustering of their activation patterns. The time series of 4000 data points, each a 256 dimensional vector, are clustered into six cluster[2].

Ideally these clusters should group the encodings into meaningful and distinct classes. In order to test this, we now correlate the six clusters with the six most common action combinations[3]. As actions and the visual execution of them do not always match up exactly (i.e. after pressing the jump button the agent is in the air for several frames and only reaches the highest point several frames after the action was selected) we perform the correlations for a 20-frame window. The vectors which are being correlated are both binary. For the six clusters, the binary vector encodes for each frame if it belongs to a specific cluster and for the actions the vector encodes for each frame if it belongs to a specific action combination or not. This means that the correlation values represent the correlation of two binary vectors of length 4000. For the offset correlations, we shift the action vector either to the left or the right such that the cluster assignment at frame t now matches up with action at frame t+1 or t-1 respectively. This gives us the correlation between cluster assignments and actions and therefore informs us if there is structure in the visual encoding that correlates with the actions selected.

In figure 4 on the top left one can see the correlations of the action *Forward + Turn Right* with the encodings of the six clusters. The highest magnitude of correlation here is at zero offset. However, also the cluster association of the observations a few frames before and after show an increase in correlation/negative correlation. When comparing this with the encodings of an autoencoder or the random untrained network one can see that there is a clear association between the learned image encoding and the actions. The trained agent has with correlation between -0.36 and 0.8 a much bigger range than the autoencoder (-0.14, 0.2) and the random network (-0.12 - 0.14). These stronger positive and negative correlations show that frames that are assigned to one cluster are more or less likely to be associated with a certain action. As the clusters are created and assigned based on

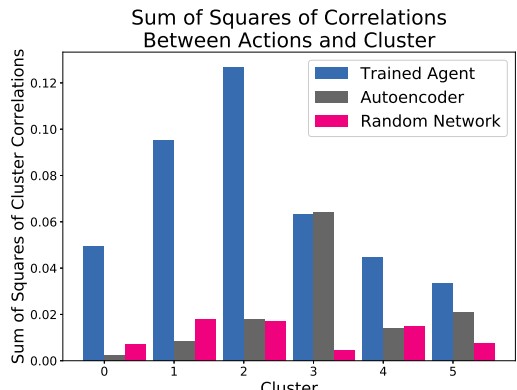

Figure 5: Sum of squares of the correlations of action combinations for each cluster. One can see that the trained agent has more expressive and distinct correlations with actions in the different clusters than a random network or the autoencoder.

the activations in the visual encoding, this means that there is a connection between distinct activation patterns and actions. Also, the activation patterns in the embodied agent preceding and following an action contain some information about it. The same correlation increase of clusters in a trained agent can be seen when looking at the correlation of clusters with semantic image content such as level doors (see figure 6). These results show that the learned representations of the visual input encode semantically meaningful and action relevant information.

The bottom part of figure 4 shows the average correlation in a 10-frame window for all six action combinations. One can see that every cluster has a unique combination of correlations or anti-correlations with the different action combinations. Some action combinations have a specific cluster which seems to mostly represent this action. In the random network in contrast, we can not find correlations of high magnitude or distinctive features between the cluster. The autoencoder has some clusters with higher correlations. However, when looking at the the correlations over time one can see that these are more long time correlations as opposed to the precisely times correlations in the embodied agent. This may be a side effect of the trained agents policy who for example tends to perform the action 'Forward, Camera Left, Turn Right' (grey) more often in puzzle rooms which might be represented uniquely in the autoencoder due to the purple colors but isn't necessarily any kind of action encoding. Due to the more long term correlations this side effect seems to be the more likely explanation.

---

[2]The choice for the number of cluster was made after comparing the inter cluster variance and silhouette score for different numbers of cluster (see appendix, figure 11)

[3]For the selection of action combinations and their distribution see appendix (figure 10, right)

Figure 5 makes this difference in correlation and therefore meaningfulness of clusters more apparent. When calculating the sum of squares of the correlations in each cluster the trained agent outperforms the random network in every cluster and the the autoencoder only marginally outperforms the trained agent in one of the clusters. The overall sum of squared correlations for the trained agent (0.41) is much higher than the one for the autoencoder (0.13) and the random network (0.07). This shows that the learned encoding has a structure which correlates with the actions as well as the image content which is impressive given the dimensionality of the observations ($\sim$84.000).

Figure 6: Correlation of clusters with Doors. Comparison of trained agent (max=0.42, min=-0.11), autoencoder (max=0.15, min=-0.09) and random network (max=0.09, min=-0.08).

## 3.5 Conceptual Similarities, Generalization and Robustness

To visualize the encodings and to investigate how conceptually similar inputs are represented, we project the activations in the visual part of the embedding layer into a two-dimensional space using t-SNE (Maaten & Hinton, 2008). Figure 7 shows the 4000 encoding activations projected into this 2D space, colored by the corresponding action combination[4]. Additionally, all data points where the visual observation contained a level door are circled in red. This makes it possible to look at the spatial arrangement of encodings in high dimensional space with respect to semantic and action-oriented content.

Even in the two-dimensional projection of the data, one can see a very good separation between points associated with the different actions. Also, the frames showing level doors tend to be positioned close to each other within their respective action cluster even though they show doors under very different illumination conditions (see the three example pictures in the top left of figure 7). As the network's task is not to recognize doors or other objects, but to navigate in a 3D world, it is important to encode the visual information in this way. A door in the right part of the frame needs to be encoded differently than a door in the center or left part of the frame. However, two doors in the right part of the frame under different illumination should be encoded very similarly. This meaningful and action relevant way of encoding the input can be seen in figure 7[5]. It shows that in the visual encoding of the input conceptually similar images are positioned close to each other, giving low importance to perceptual similarities. This means that the network encodes the input in an action-oriented way (Clark, 1998) and is rather invariant towards irrelevant parts of the input such as illumination or texture.

To investigate if conceptually similar input images also lay close to each other in high dimensional space, we can calculate the inter-class distance and variance of encodings associated with the different action combinations. Table 1 lists the distances and variances between encodings belonging to the same action combination divided by the overall distances and variance in the data.

In the trained agent, both distance and variance reduce strongly between the first four action combinations (0.78 and 0.58 respectively). In the autoencoder and the random network, there is no change

---

[4]For an interactive visualization to explore the observations associated with each point see here.

[5]As a control that the structure does not simply originate from the statistics of the input images, t-SNE on the activations of the autoencoder and the untrained agent on the same input images are shown in figure 12 in the appendix.

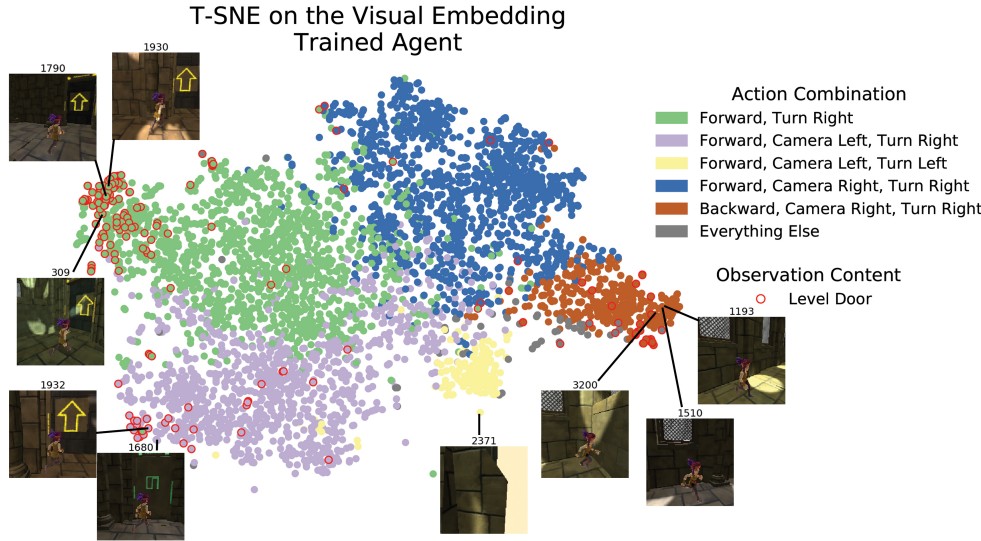

Figure 7: T-SNE performed on the visual embedding of a trained agent colored by action combination associated with each frame. Images show example agent observations which created the encoding activation associated with the point it is connected to. Encodings associated with frames containing level doors are circled in red.

in the distance or variance when comparing all data points to points belonging to one action combination. The last two action combinations which represent backwards motion and all other rare action combinations actually have an increased variance in the trained agent. This may be due to a very variable use of backwards motion when seeing possibly confusing visual input and the accumulation of multiple action combinations (also including jumping) in the last action category. However, we can see that at least for the first four action combinations the encodings of conceptually similar input frames are also closer together in the high dimensional space (256 dimensions) of the visual encoding. Also when comparing the absolute change in variance and distance for each action cluster (last row in table 1) the trained agent outperforms the two other networks strongly indicating a very distinct action encoding in the network trained in an embodied setup.

Table 1: Average distance and variance within points belonging to the same action combination divided by the overall distance or variance between all encodings respectively.

| Fraction of overall dist/var | Trained Agent | | Autoencoder | | Random Network | |
| --- | --- | --- | --- | --- | --- | --- |
| | Distance | Variance | Distance | Variance | Distance | Variance |
| Average | **0.96** | **0.94** | 1.06 | 1.12 | 1.07 | 1.16 |
| Action 1-4 | **0.78** | **0.58** | 1.02 | 1.04 | 1.03 | 1.06 |
| Action 5-6 | 1.32 | 1.66 | **1.12** | **1.28** | 1.15 | 1.35 |
| Mean(Abs(1-d/v_a)) | **0.25** | **0.5** | 0.09 | 0.18 | 0.1 | 0.23 |

## 4 CONCLUSION

The results presented in this paper show that a neural network, trained in an embodied framework, can learn stable and meaningful representations of its high dimensional input. Compared to an autoencoder, the representations of the embodied agent are encoded in a sparse and efficient way and better reflect information encoding found in animals. The information encoded in the latent representation of the network is mainly action focused, but also contains general concepts of action relevant objects such as doors, disregarding irrelevant information such as illumination. Overall these results suggest deep reinforcement learning as a promising method for investigating stable representation learning similar to what is known from biological findings.

AUTHOR CONTRIBUTIONS

X.X. conceived the idea to investigate latent representations in deep reinforcement learning agents, trained the agents, designed the experiments, analyzed the data and took the lead in writing the manuscript. All authors discussed the experiment design and results and contributed to the final manuscript. X.X, X.X. and X.X. supervised the project.

ACKNOWLEDGMENTS

The project was financed by the funds of a research training group provided by the XXX (ID XXX).

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

## A  APPENDIX

### A.1  TRAINING SPECIFICS

| | |
|---|---|
| Batch Size | 256 |
| Beta | 5.0e-3 |
| Buffer Size | 1024 |
| Epsilon | 0.2 |
| Gamma | 0.999 |
| Lambd | 0.9 |
| Learning Rate | 1.0e-4 |
| Normalize | True |
| Use Recurrency | False |
| # Parallel Environments | 16 |

### A.2 AUTOENCODER

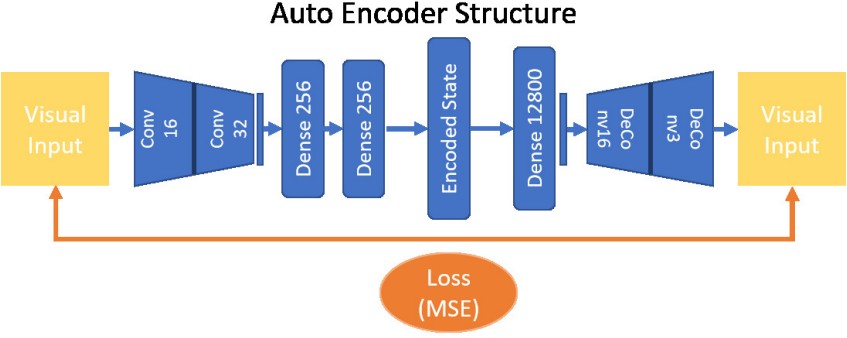

Figure 8: Network structure of the autoencoder.

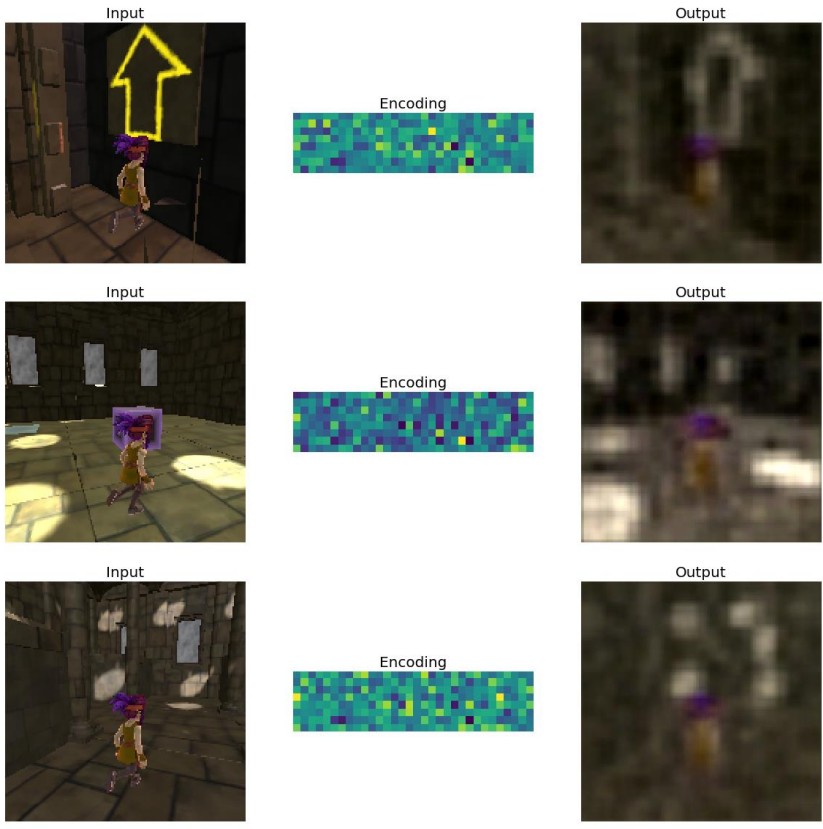

Figure 9: Autoencoder example input, encoding and output.

## A.3 ACTIONS

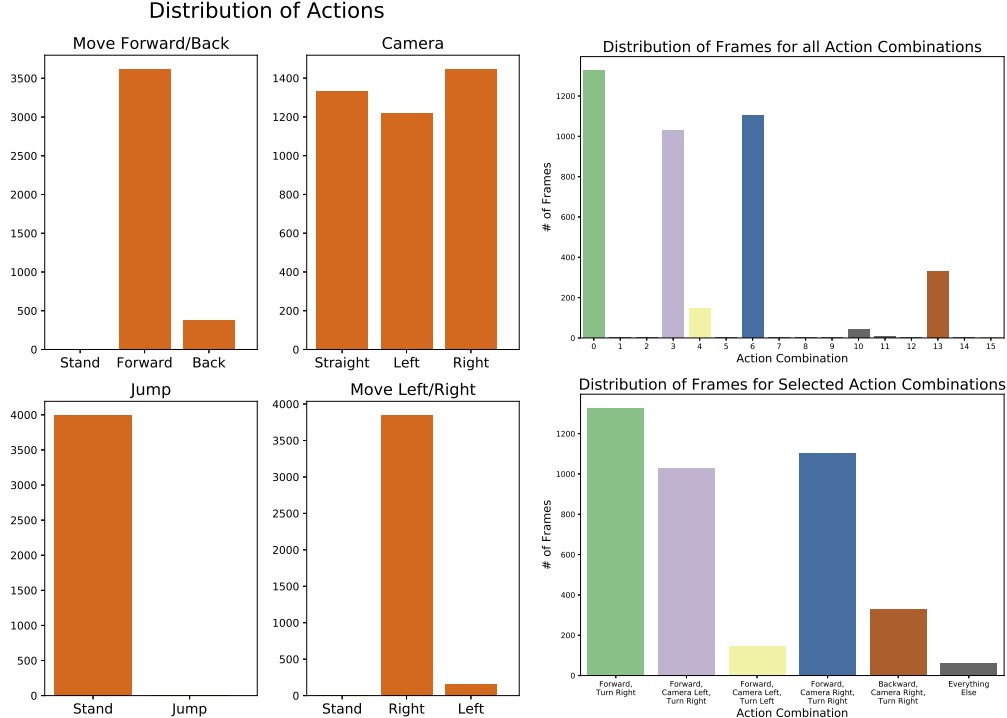

Figure 10: Distribution of actions over one inference run of 4000 frames. (Left) Distribution in the four action branches. (Top Right) Distribution of all existing action combinations. (Bottom Right) Distribution of all action combinations used in the paper. All actions marked in grey in the plot above are aggregated in the last action combination.

## A.4 K-MEANS CLUSTERINGS

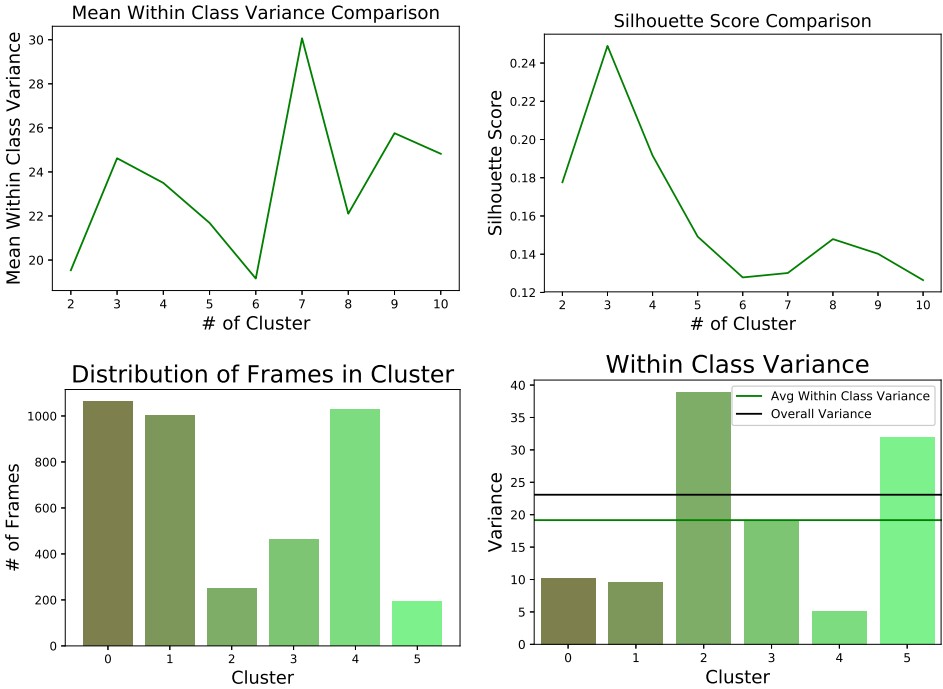

Figure 11: (Top) Analysis used for the choice of number of clusters in section 3.4. Both mean within class variance as well as the silhouette score for different number of clusters have their minimum at six clusters. (Bottom) Statistics of the k-means clustering with six clusters.

## A.5 T-SNE

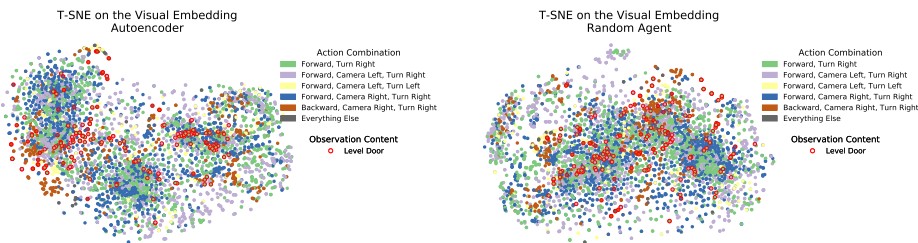

Figure 12: T-SNE performed on the visual embedding of a trained autoencoder (left) and a random network (right) colored by action combination associated with each frame. Frames where the observation contained a level door are circled red.

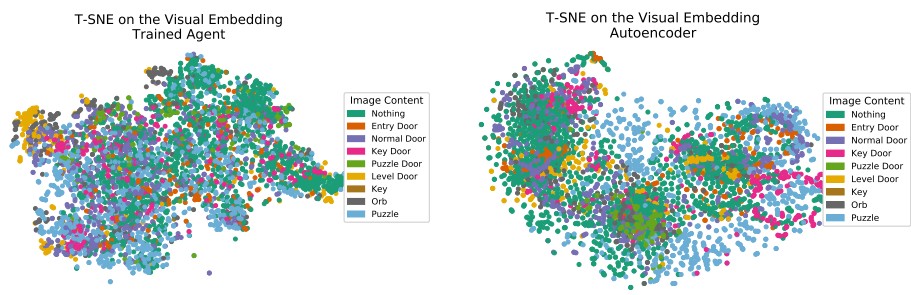

Figure 13: T-SNE performed on the visual embedding of a trained Agent (left) and a trained autoencoder (right) colored by semantic content of the visual observation associated with the encoding.

## A.6 ANALYSIS OF HIGH-D SPACE

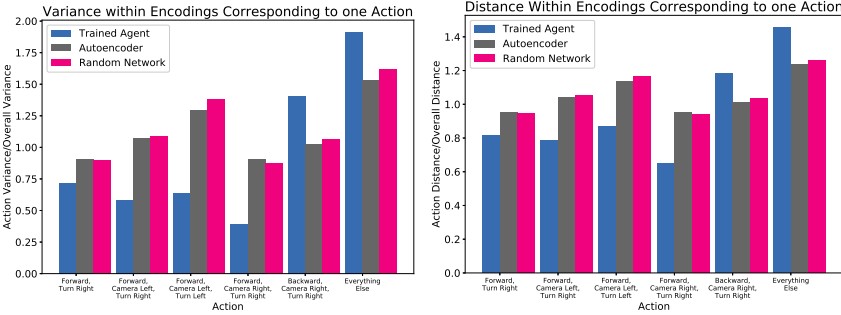

Figure 14: Distances and variances within the encodings for individual actions in the original high dimensions encoding space (dim=256). Data in numbers is also given in 1 accumulated over actions.

