# OpenReview forum: "Learning Semantically Meaningful Representations Through Embodiment"
_ICLR.cc/2020/Conference — Reject_

### Official Review · AnonReviewer1 · 2019-10-12
**Official Blind Review #1**

**Rating:** 1

**Review:**


Paper Summary: The goal of the paper is to analyze what information is encoded in the representation learned using RL for a specific game. The paper shows that the activations are sparse and the activation patterns are distinct and shows that the conceptually similar images are clustered together in t-sne visualization.

Paper Strengths:

The paper starts with a nice introduction that embodiment is useful for perception. However, the main content of the paper is very different from the introduction.

Paper Weaknesses:

The conclusions of the paper are either already known or very trivial. So there is nothing new for the community to benefit from. Please refer to my comments below for more information.

The conclusion of section 3.3 is that "the agent learns to use sparse activation patterns and even leaves some of the available neurons completely unused". This has nothing to do with embodiment. The same patterns are observed in non-dynamic tasks such as image classification. The CNNs are usually over-parameterized.

The conclusion of section 3.4 is that "When comparing this with the encodings of a random untrained agent one can see that there is a clear association between the learned image encoding and the actions". That is the whole point of training. We train the models to find correlations between actions and observations. It is obvious that there is more correlation compared to a random agent.

It is shown in section 3.5 that similar images will be next to each other in the t-sne visualization. It is obvious that this happens.

Due to the issues mentioned above, I do not think there is anything new in the paper and I vote for rejection.



**Experience Assessment:**

I have published in this field for several years.

**Review Assessment: Checking Correctness Of Derivations And Theory:**

N/A

**Review Assessment: Checking Correctness Of Experiments:**

I carefully checked the experiments.

**Review Assessment: Thoroughness In Paper Reading:**

I read the paper thoroughly.

---

> ### Author Response · Authors · 2019-11-08
> **Reviewer #1 - Reply**
>
> Dear Reviewer #1,
>
> Thank you for your very insightful feedback. We greatly appreciate your concerns about the paper and have thought a lot about how to better demonstrate the strength of the results and their novelty. Based on your comments we decided to take these actions:
>
> -	We agree that the comparison to a random agent is a rather soft baseline. Therefore, we now include further analysis comparing the agent trained in an embodied set up to an auto encoder trained on the same visual stimuli. The results from this analysis show that the representations of the visual input in the embodied agent are very different to the representation learned by the auto encoder. There is a much stronger encoding of actions in the embodied agent and no more action encoding in the trained action encoder than in the random network.
>
> -	This observation might be considered as obvious. However, the difference of representations in the visual module under different training conditions is an important factor when analyzing representation learning. We think that this insight is important to keep in mind when studying the brain and the mechanisms of what kind of representations are learned under which conditions are interesting to study. As the learning conditions of an embodied network are closer to the ones of humans and animals we think it is interesting to investigate these representations further.
>
> -	Your concern for section 3.3 that many CNNs are over-parameterized, regardless of embodiment, will be addressed by the addition of the results from the trained auto encoder. It is true that many CNNs trained in a supervised setup are over-parameterized, however, those CNNs are usually bigger than two layers and often have input of a smaller dimensionally. In our setup the input has dimensionality 84.672 and is compressed to a vector of size 256 which is only about 0.3% of the original input size. When training the autoencoder with a network of the same structure it learns no sparse representations of the input and uses most neurons in 100% of the input frames with varying strengths in the activations while the embodied network activates its neurons much more selectively.
>
> -	In your last comment you point out that it is obvious that similar images will be next to each other in the t-SNE visualization. We apologize for this misunderstanding. It addresses the core message of the manuscript that conceptually similar images, not perceptually similar images, are close in the representational space. This is the very advantage of reinforcement learning by the embodied agent. We will revise and precisely phrase the respective sections and figure 7. When looking at the representations of the auto encoder such conceptual similarities are not as strongly encoded, and focus is more on pure perceptual similarities such as overall illumination.
>
> After adding these changes and additional results we would be very happy if you could take another look and evaluate our results under the light that this is the first step of a bigger project investigating the relationship between training environment, training conditions, and learned representations. Even though this paper alone might not be ground breaking, it lays the foundations of many more results to come investigating no reward tasks using curiosity, hierarchical representations and different learning schemes to make network training more biologically inspired.
>
> Sincerely,
> the authors

---

### Official Review · AnonReviewer3 · 2019-10-22
**Official Blind Review #3**

**Rating:** 3

**Review:**

This work builds on the embodied cognition literature, hypothesizing that representations learned in embodied agents will be of improved “quality” compared to non-embodied models, such as neural networks trained on static supervised datasets.

The authors provide an excellent motivation to the work, with the introduction nicely laying the groundwork for their hypothesis. In general the motivation is quite strong, and research along these directions will no doubt be of value to the field.

To assess their hypothesis, the authors compare the representations learned in trained vs. random agents on the Unity Obstacle Tower Challenge, and demonstrate that trained agents develop semantically meaningful, sparse representations, without explicit regularization.

Unfortunately, the work doesn’t provide adequate baselines to properly assess the hypothesis. With the data presented, we can only make claims about *trained* vs. *untrained* agents (both of which are embodied!). In other words, an embodied agent with a random policy is not equivalent to a non-embodied agent. Thus, the data only support the conclusion that performance and task-relevant policies drive good representation learning in embodied agents, which is altogether not surprising, as one wouldn’t expect representations to be good in a randomly initialized network.

To assess wither *embodiment* is a critical factor for driving good representations, the authors should compare to a model that learns from a static supervised dataset. Curiously, this idea is alluded to in the introduction, but not followed up on.

Overall, there are some nicely presented ideas but the work is unfortunately incomplete, and the results cannot support the hypothesis laid out.

As a final note, the authors are encouraged to remove any assignments of gender to the agent (“he/him” is unnecessarily used throughout).


**Experience Assessment:**

I have published one or two papers in this area.

**Review Assessment: Checking Correctness Of Derivations And Theory:**

N/A

**Review Assessment: Checking Correctness Of Experiments:**

I assessed the sensibility of the experiments.

**Review Assessment: Thoroughness In Paper Reading:**

I read the paper at least twice and used my best judgement in assessing the paper.

---

> ### Author Response · Authors · 2019-11-08
> **Reviewer #3 - Reply**
>
> Dear reviewer #3,
>
> We are very grateful for your comments and feedback on the paper and will work them into the revised paper. Originally the random agent presented itself as a good comparison as we could use the exact same network structure, input and representation dimensionality. However, we understand that this is not the most convincing comparison we can make and have already trained an auto encoder on the static observations collected by the embodied agent, using the same network structure up to the encoding layer. We will include these results in the revised version of the paper and contrast them with the representations learned by the embodied agent. We hope that these additional results will help us make a stronger point than the comparison to a random agent.
>
> Additionally, we apologize for the use of he/him to refer to the agent. This was not supposed to be any kind of politically statement but was a simple mistranslation from the authors native tongue where the noun for agent is grammatically male. We will of course change this in the revised version.
>
> Sincerely,
> the authors

---

> > ### Comment · AnonReviewer3 · 2019-11-14
> > **Thanks**
> >
> > Thank you for the corrections and for the additional experiment. Unfortunately I think there's a confounding factor with the auto-encoder experiment in that the autoencoder objective (and presumably the network architecture) explicitly encourages a dense representation. This makes it difficult to isolate embodiment as the critical differentiating factor. Sadly I do not have a concrete suggestion regarding an appropriate baseline, and I think it will require a significant amount of thought and effort. But setting the correct baseline is the only way to validate the claims about embodiment, and is the only route to showing its importance. Unfortunately I cannot recommend publication at this time, but I encourage the authors to explore these issues further as they are definitely important to the field.

---

> > > ### Author Response · Authors · 2019-11-15
> > > **Regarding concerns**
> > >
> > > We agree with the concerns regarding the objective optimized by the autoencoder. However, it is clearly a better comparison than the previous baseline. Further, as the reviewer points out, there is no canonical choice of a baseline for comparison. Therefore, we consider the comparison to the auto encoder as valuable. We phrased our conclusions carefully and point out that they are obtained by comparison to the autoencoder. In this way we think that our claims are carefully and correctly phrased.
> > >
> > > In addition, we would like to state that obviously any setup different from the embodied agent setup will enforce different types of representations. This is unavoidable and one of the main points of the paper. We like to show that the way a network is trained has a huge impact on the kind of representations learned and that the representations learned with a closed loop between action and perception are closer to representations found in nature than the ones learned in other classical training setups without dense semantic labeling such as an auto encoder. We therefore do not agree that that the autoencoder enforcing a denser representation is a reason for rejection as it is a key fact that we are presenting and part of the main message. We will try to formulate this more clearly in the paper.
> > >
> > > We agree that there are other setups out there that would isolate embodiment more clearly and we will explore this further in the future. However, we think that already the representation analysis in itself and the comparison to a standard architecture like the autoencoder present very valuable insights, motivating many further research directions.
> > >
> > > Sincerely,
> > > The authors

---

### Official Review · AnonReviewer2 · 2019-10-23
**Official Blind Review #2**

**Rating:** 3

**Review:**

Paper summary:

This is an empirical study of the representations learned by a reinforcement learning agent. An agent is trained, using a standard RL algorithm, to solve puzzles by navigating through a 3D visual environment (Unity obstacle tower challenge). The analyses in the paper show that the visual representations of the trained agents are sparse and cluster according to the actions performed by the agent. The goal of the paper is to show that these features are due to the embodied nature of the agent. Specifically, the paper states that “the quality of the representations learned shows the strength of embodied learning and its advantages over fully supervised approaches with regards to robustness and generalizability”.

Decision:

I suggest to reject this paper. While the topic is interesting and the paper is clearly written, there is a lack of control/comparison experiments, such that the paper’s conclusions are not backed up by the analyses. However, with more experiments, I think this line of research has large potential.

Further justification for the decision:

My main criticism is that the paper claims to show that embodiment is important for representation learning, but never actually compares representations learned by an embodied agent to representations learned in some other way.

Comparing to a random network is a sanity check, but not sufficient to support the paper’s claims.

One way to experimentally dissociate embodiment/agency from supervised learning would be to train two separate models, one that is “embodied”/active, and another that gets the same sensory input, but without embodiment (“passive”). The passive network could be trained using the sensory inputs recorded while training the active network. The passive network could be trained to predict the value and/or the actions of the active network. Thus, the passive network would be trained in a supervised way, whereas the active network would be trained by RL (i.e. in an embodied way).

The representations of the two networks could then be compared using the analyses used in the paper. This setup would experimentally isolate the effect of embodiment. Without such comparisons, it is unclear whether representations learned in a supervised way would be any different from those learned by RL.

In addition to the descriptive analyses presented in the paper, a transfer learning approach could test whether there is actually a functional difference between the “embodied” and the supervised representations: Take the representations of the “active agent” and the “passive agent” and freeze the weights. Then re-initialize and re-train only the dense layer before the action probabilities on the RL task, leaving everything else frozen. Does the model from the “active agent” do better on the RL task? This would suggest that the embodied agent learned better representations.

Minor comment: The website contains a link to a YouTube profile that is not completely anonymous (it contains the first name and a profile photo). While I did not identify the authors when visiting the paper website, I recommend removing links to personal YouTube profiles and create submission-specific anonymous accounts.

**Experience Assessment:**

I have read many papers in this area.

**Review Assessment: Checking Correctness Of Derivations And Theory:**

N/A

**Review Assessment: Checking Correctness Of Experiments:**

I carefully checked the experiments.

**Review Assessment: Thoroughness In Paper Reading:**

I read the paper at least twice and used my best judgement in assessing the paper.

---

> ### Author Response · Authors · 2019-11-08
> **Reviewer #2 - Reply**
>
> Dear reviewer #2,
>
> Thank you very much for your valuable input. We appreciate your ideas for further experiments very much and think that both experiments you propose are very interesting. We will definitely run the controlled experiment isolating embodiment that you suggested as well as the experiment using the two different learned representations in the RL task and comparing them.
> However, seeing the tight deadline of the paper submission and the fact that the training of these agents can take up to a month on our available hardware we propose to add a slightly modified control experiment from which we already have results. We have trained an autoencoder with the exact same network structure before the embedded layer on the visual input collected by the embodied agent. We will add the analysis of the representations learned in the autoencoder and compare them to the representations of the embodied agent. We think that this comparison will make a much stronger case for the action oriented and robust encoding learned in the embodied setup compared to the non-embodied setup.
>
> We have also removed the video link from the website.
>
> Sincerely,
> the authors

---

> > ### Comment · AnonReviewer2 · 2019-11-15
> > **Response**
> >
> > I do not think that the autoencoder experiments are a useful comparison because the autoencoder is trained on a completely different task than the RL agent: image reconstruction vs. value/action prediction. The autoencoder is trained to encode a lot of information that is irrelevant to the RL agent.
> >
> > Instead of training an autoencoder on image reconstruction, it would have been more informative to train a supervised version of the network to predict values and actions generated by the RL agent. This would have a similar computational cost (probably less) than training the autoencoder, so resource limitations cannot have prevented this (no re-training of the RL agent would have been necessary).
> >
> > I agree with Reviewer #3 that this work is not ready for publication at this point, but that the authors should be encouraged to continue with this important research as suggested.

---

> > > ### Author Response · Authors · 2020-01-06
> > > **Justification for the Experiment Used**
> > >
> > > Dear Reviewer 2,
> > >
> > > the reason that we incorporated the results of an auto encoder instead of your proposed solution is not that we are incapable of understanding or implementing your suggestion but, as already said in the previous comment, that we could not perform this analysis during the two week review period. I was on my honeymoon during this time and did not have access to hardware capable of training any kind of new setup. If you didn't like the idea of the auto encoder it would have been nice to know that soon after I suggested it, before I waste days of my honeymoon stressing about adapting the paper to it. It would have also been nice to receive your comment less close to the deadline so we have time to respond to it and justify our decision. We chose the auto encoder as it is a very commonly used way of training a neural network in an unsupervised fashion and we wanted to point out the difference of our approach to other commonly used approaches. Of course we will also perform an analysis on your suggested training set up and I will update the paper as soon as I have the results.

---

### Decision · Program_Chairs · 2019-12-19

**Decision:**

Reject

**Comment:**

What is investigated is what kind of representations are formed by embodied agents; it is argued that these are different than from non-embodied arguments. This is an interesting question related to foundational AI and Alife questions, such as the symbol grounding problem. Unfortunately, the empirical investigations are insufficient. In particular, there is no comparison with a non-embodied control condition. The reviewers point this out, and the authors propose a different control condition, which unfortunately is not sufficient to test the hypothesis.

This paper should be rejected in its current form, but the question is interesting and hopefully the authors will do the missing experiments and submit a new version of the paper.